# Exosomes Derived from Yak Follicular Fluid Increase 2-Hydroxyestradiol Secretion by Activating Autophagy in Cumulus Cells

**DOI:** 10.3390/ani12223174

**Published:** 2022-11-16

**Authors:** Ruihua Xu, Jinglei Wang, Meng Wang, Liqing Gao, Rui Zhang, Ling Zhao, Bin Liu, Xiaohong Han, Abdul Rasheed Baloch, Yan Cui, Sijiu Yu, Yangyang Pan

**Affiliations:** 1College of Veterinary Medicine, Gansu Agricultural University, Lanzhou 730070, China; 2Dr. Panjwani Center for Molecular Medicine and Drug Research, International Center for Chemical and Biological Sciences, University of Karachi, Karachi 75270, Pakistan; 3Gansu Province Livestock Embryo Engineering Research Center, Lanzhou 730070, China

**Keywords:** yak follicular fluid exosomes, autophagy, 2-hydroxyestrondiol, cumulus cells

## Abstract

**Simple Summary:**

Exosomes in the follicular fluid can carry and transfer regulatory molecules to recipient cells, thus influencing their biological functions. In this study, after being treated with yak follicular fluid for 24 h, most yak cumulus cells took up the exosomes, activated autophagy, and increased their secretion of 2-hydroxyestrodiol. Conversely, the inhibition of autophagy with the chemical reagent 3-methyladenine blocked these effects, suggesting that autophagy has an important role in 2-hydroxyestrodiol secretion in the yak cumulus cells. The results of rapamycin treatment of the yak cumulus cells showed a similar effect on yak follicular fluid exosomes as there was an increase in 2-hydroxyestrodiol secretion due to the activation of autophagy. The results suggest that autophagy is activated by yak follicular fluid exosomes and this increases 2-hydroxyestrodiol secretion in the yak cumulus cells.

**Abstract:**

Exosomes in the follicular fluid can carry and transfer regulatory molecules to recipient cells, thus influencing their biological functions. However, the specific effects of yak follicular fluid exosomes on 2-hydroxyestrodiol (2-OHE_2_) secretion remain unknown. Here, we investigated whether yak follicular fluid exosomes can increase 2-OHE_2_ secretion through the activation of autophagy in cumulus cells (YCCs). In vitro cultured YCCs were treated with yak follicular fluid exosomes for 6, 12, and 24 h. The effects of yak follicular fluid exosomes on autophagy and 2-OHE_2_ secretion were evaluated through real-time quantitative fluorescence PCR (RT-qPCR), Western blotting (WB), transfected with RFP-GFP-LC3, immunohistochemistry, and ELISA. To further investigate whether 2-OHE_2_ secretion was related to autophagy, YCCs were administered with yak follicular fluid exosomes, 3-methyladenine (3-MA), and rapamycin (RAPA). The results revealed that treatment with yak follicular fluid exosomes activated autophagy in YCCs and increased 2-OHE_2_ secretion. Conversely, the inhibition of autophagy with 3-MA blocked these effects, suggesting that autophagy has an important role in 2-OHE_2_ secretion in YCCs. Treatment of YCCs with rapamycin showed similar results with yak follicular fluid exosomes as there was an increase in 2-OHE2 secretion due to the activation of autophagy in the treated cumulus cells. Our results demonstrate that autophagy is enhanced by yak follicular fluid exosomes, and this is associated with an increase in 2-OHE_2_ secretion in YCCs.

## 1. Introduction

Extracellular vesicles (EVs) are cell-membrane-derived structures found in biofluids. Exosomes are EVs derived from the endocytic cell compartment after the fusion of multivesicular bodies with the plasma membrane [1]. Exosomes, which have a diameter of 30–150 nm, are important for paracellular secretion and intercellular communication [2]. They can carry and transfer regulatory molecules, such as coding RNAs, noncoding RNAs, DNA, proteins, growth factors, and lipids, and transfer cytosolic macromolecules to target cells to induce alterations in their physiological functions [3,4]. In addition, exosomes may mediate intercellular communications between cells and tissues. Exosomes have been isolated from a wide variety of biofluids, including plasma, urine, uterine luminal fluid, and, most relevant to this investigation, follicular fluids [3].

Macroautophagy/autophagy is a lysosomal-dependent self-degradation process that acts as a cell survival mechanism for the disassembly of superfluous or damaged cell cytoplasmic components and the recycling of bioenergetic molecules [5]. During autophagy, autophagosome initiation, elongation, and maturation, fusion of the autophagosomes to the lysosomes, and degradation within the lysosomes are tightly governed by multiple signaling mechanisms [6,7]. In mammals, the autophagy process correlates with a wide range of physiological events, such as embryonic development, responses to stress, and disease progress [8]. Accordingly, autophagy has become a fascination in the field of biological science, including the regulatory effects of yak follicular fluid EVs on the autophagy of cumulus cell culture in vitro. In recent years, numerous studies have revealed the molecular mechanisms in exosome and autophagy interactions, and identified that the mTOR [1], toll-like receptor, and STAT3/Bcl-2 signaling pathways are important for exosome regulation of autophagy [9,10]. Recent studies have shown that exosomes, and their cargo can regulate autophagic pathways via different mechanisms. For example, exosomes released from hypoxic cardiomyocytes inhibit autophagy by transferring miR-30a in a paracrine manner [11].

Estrogens are signaling molecules converted from cholesterol to bioactive steroid hormones and this process is catalyzed by a series of different enzymes such as cytochrome P450 (CYP), the hydroxysteroid dehydrogenase (HSD) family, some lipid-binding proteins, and transporters [12,13]. 17β-estradiol mediates several cellular effects through its non-estrogenic metabolites (particularly 2-hydroxyestradiol and 4-hydroxyestradiol), which result after oxidative metabolism [14]. In this regard, the 17β-estradiol C2 hydroxylation leads to the formation of 2-OHE_2_, a metabolite that has no uterotropic activity, with a low affinity for estrogen receptors. It is reported that 2-OHE_2_ may provide a renoprotective effect, which is not mediated by estrogen receptors. Furthermore 2-OHE_2_ can inhibit DNA synthesis and cell proliferation and has a beneficial effect on the meiotic and developmental competence of the enclosed oocytes in rat GMC in culture and can improve oocyte developmental competence [15].

Yaks (B. grunniens), which are distributed in the Qinghaie-Tibet Plateau and its adjacent alpine or subalpine regions, are important animals for local herdsmen and have irreplaceable social, ecological, and economic importance in the plateau area [16,17]. However, harsh, oxygen-limited, and high-altitude environmental characteristics result in low yak estrus rates and fecundity [18]. Therefore, the wider application and promotion of assisted reproductive technology (ART) in yaks will help breeding strategies to help preserve the health of yak populations for future generations [19]. Cumulus cells are the main site of ovarian steroid secretion and the barrier between oocytes and follicular fluid, which participate in oocyte growth, meiosis, ovulation, and fertilization in mammals [20,21]. Yak follicular fluid exosomes can affect the biological functions of cumulus cells, but the specific effects of 2-OHE_2_ secretion remain unknown. Therefore, we hypothesized that in a monoculture system, yak follicular fluid-derived exosomes could affect autophagy and 2-OHE_2_ secretion of YCCs. To test this working hypothesis, we analyzed the effects of yak follicular fluid exosomes on cumulus cell autophagy and 2-OHE_2_ secretion, and further addressed the relationship between autophagy and 2-OHE_2_ secretion. The results will help for analysis of the reproductive physiology of yak ovaries and provide a reference for the development of new strategies to improve oocyte quality and the efficiency of invitro maturation in yaks and other mammals.

## 2. Materials and Methods

### 2.1. Preparation of Yak Ovaries and Follicular Fluid

Ovaries were collected from a slaughterhouse located in Xining, Qinghai. Then, the ovaries were transported to the laboratory at 25–30 °C in sterile saline containing 1% penicillin and streptomycin for no more than 4 h.

The ovaries were repeatedly flushed three times with normal saline until they were blood free. The follicular fluid from mature follicles (>8 mm in diameter) was withdrawn with a syringe and then stored at −80 °C in a 50 mL centrifuge tube.

### 2.2. Isolation and Identification of Exosomes

The collected follicular fluid was divided into two equal volumes for exosome isolation by centrifugation. The cells in the yak follicular fluid were removed by centrifugation at 500× *g* for 15 min (Beckman Coulter, Brea, CA, USA), followed by centrifugation at 12,000× *g* for 15 min to remove the microvesicles. The supernatant was then collected and centrifuged at 100,000× *g* at 4 °C for 90 min. After centrifugation, the supernatant was discarded, and the exosome resuspended in 1× PBS and washed once by centrifugation at 100,000× *g* at 4 °C for 90 min. Some of the exosomes were resuspended in 3 mL sterile PBS for identification, and the other exosomes were resuspended in 3 mL DMEM/F-12 and stored at −80 °C for further experimentation. Meanwhile, transmission electron microscopy (TEM, HT-7800, Hitachi Ltd., Tokyo, Japan) was used to observe the morphology of the yak follicular fluid-derived exosomes. Nanoparticle tracking analysis (NTA, NanoFCM Co., Ltd., Nottingham, UK) was performed to detect the size distribution and concentration of the exosomes. The expression of the exosome protein markers TSG101 and CD63 was analyzed using Western blot.

### 2.3. Isolation and Culture of Yak Cumulus Cells

After the ovaries were washed three times with sterile saline containing 100 IU/mL penicillin and 100 mg/mL streptomycin sulfate, cumulus–oocyte complexes (COCs) were aspirated from the healthy antral follicles [22]. They were then transferred to a petri dish. After maturation culture in vitro, COCs were then transferred to 1% hyaluronidase for repeated light pipetting until the cumulus cells fell off. The hyaluronidase was neutralized with DMEM/F-12 containing 10% FBS and centrifuged at 300× *g* for 10 min. The cell pellet was then resuspended with 5 mL DMEM/F-12 medium containing 10% FBS and antibiotics. Finally, the cells were cultured in 25 cm^2^ cell culture flasks in a cell incubator. After fusion growth of the primary cells, some of the cells were frozen and the others were passaged at a ratio of 1:3, and only primary cells or those that were passaged twice were used for subsequent experiments. Cell growth was observed and photographed using an inverted phase contrast microscope (Olympus CK41, Tokyo, Japan).

### 2.4. Exosome Labeling and Co-Incubation with YCCs

Yak follicular fluid exosomes were labeled with a PKH-26 Red Fluorescent Cell Linker Mini Kit (MINI26-1KT, Sigma-Aldrich, St. Louis, MO, USA). Exosome pellets were resuspended in 1 mL Diluent C. Then, 6 μL PKH-26 was added to 1 mL Diluent C and incubated for 5 min. This was stopped by the addition of an equal volume of 10% bovine serum albumin (BSA). Labeled exosomes were ultracentrifuged at 100,000× *g* for 90 min at 4 °C, washed with sterile PBS, and ultracentrifuged again. Finally, PKH26-labeled exosomes were resuspended in DMEM/F-12 prior to the uptake assay.

Cumulus cells were seeded into cell culture dishes (35 mm) at a density of 3 × 10^6^ cells/dish, grown to 60% confluence. They were then starved with DMEM/F-12 for 12 h. For the control group, no exosomes were added. For the experimental group, the cells were treated with yak follicular fluid exosomes in 5% CO_2_ and 95% air at 37 °C (1.15 × 10^7^ particles/mL). After co-incubation for 6, 12, and 24 h, the treated cells were used in the following experiments.

### 2.5. Immunofluorescence

YCCs grown on coverslips were washed with PBS twice and fixed in 2% paraformaldehyde solution for 30 min at room temperature. The cells were then permeabilized in 0.1% Triton X-100/PBS for 10 min at room temperature, washed with 0.1% Tween 20 in PBS (PBST) 5 times, blocked with 10% BSA at 25 °C for 1 h, washed with 0.1% PBST 5 times, and the primary antibodies CYP19A1 (1:250, Affinity, AF5229, Milwaukee, WI, USA), CYP1A1 (1:250, Affinity, AF5312), and beta tubulin (1:300, Bioss, bsm-33041M, Woburn, MA, USA) were separately added and incubated for 12 h at 4 °C. The cells were then stained with anti-rabbit IgG Fab2 (1:1000, Cell Signaling, 8889S, Danvers, MA, USA) and anti-mouse IgG Fab2 (1:1000, Cell Signaling, 4408S) for 2 h in the dark at room temperature and washed with 0.1% PBST 5 times and 4′,6-diamidine-2′,2-phenylindole dihydrochloride (DAPI) for 4 min. Finally, we observed and photographed the cells using fluorescence microscopy.

### 2.6. Treatment of Cells with 3-MA and RAPA

Cumulus cells were seeded into 6-well plates at a density of 3 × 10^5^ cells/dish, grown to 60% confluence, and then starved with DMEM/F-12 for 12 h. The autophagy inhibition group of cells were then treated with 3-MA (7.5 mM/mL) and the autophagy activation group were treated with agonist RAPA (12.5 and 25 μM/mL). For the control group, no exosomes, 3-MA, nor RAPA were added. After co-incubation for 24 h, the treated cells were used in the following experiments.

### 2.7. Quantitative RT-qPCR Analysis

The total RNA of YCCs was extracted using the TranZol kit (Invitrogen, Carlsbad, CA, USA) for 6, 12, and 24 h according to the manufacturer’s instructions. They were then reverse transcribed into complementary cDNA with the two-step reverse transcription kit (Accurate Biotechnology, Changsha, China). The mRNA levels were quantified using RT-qPCR. The relative expression of LC3, ATG5, ATG12, Beclin1, and P62 was normalized against β-actin. All reactions were run in triplicate using the SYBR Green Premix Pro Taq HS qPCR Kit (Accurate Biotechnology, Changsha, China) on a LightCycler^®^480 Instrument II (Roche, Basel, Switzerland) in a 20 μL reaction volume. The target and reference primer sequences were designed using Primer Premier 6.0 (Table 1). The relative expression of the target gene transcripts was calculated using the 2^−ΔΔCT^ method and subjected to statistical analysis using SPSS 22.0 software [23].

### 2.8. Western Blotting (WB) Analysis

YCCs and the exosome pellets in the dish were washed 3 times with sterile PBS and then lysed for 30 min on ice with RIPA lysis buffer, and the lysates were collected in tubes. Then, equal amounts of protein were separated on 10% or 15% SDS-polyacrylamide gel electrophoresis and transferred to polyvinylidene difluoride (PVDF) membranes. The membranes were blocked with 5% skim milk in Tris-buffered saline containing 0.1% Tween-20 for 3 h at room temperature, incubated overnight at 4 °C with primary antibody, and on the next day, they were incubated for 45 min at room temperature with a secondary antibody (Goat Anti-Rabbit IgG-HRP) [23]. Bands were visualized using an ECL kit (AB65623, Abcam, Cambridge, UK) and detected using the Amersham Imager 600 system (GE Healthcare Life Sciences, Marlborough, MA, USA). The density of the protein bands was quantified using Image J, and β-actin was used as an internal control.

The following primary antibodies were used in this study: LC3 (1:1000, Thermo Fisher, 14600-1-AP, Waltham, MA, USA), Beclin1 (1:2000, Novus Biologicals, NB110-87318, Centennial, CO, USA), ATG5 (1:500, Novus Biologicals, NB110-53813), P62 (1:2000, Abcam, AB96134), TSG101 (1:2000, Abcam, ab225877), CD63 (1:2000, Abcam, ab193349), CYP19A1 (1:1000, Affinity, AF5229), CYP17A1 (1:1000, Affinity, AF5210), CYP1A1 (1:1000, Affinity, AF5312), CYP1B1 (1:1000, Affinity, DF6399), and Goat Anti-Rabbit IgG-HRP (1:8000, Abmart, M21002, Shanghai, China), and β-actin (1:8000, Affinity, AF7018).

### 2.9. Evaluation of Fluorescent LC3 Puncta

Yak cumulus cells cultured on coverslips were transfected with RFP-GFP-LC3 at 50 MOI [1]. After adenovirus transduction for 24 h, for the control group, no exosomes were added. For the experimental group, the cells were treated with yak follicular fluid exosomes in 5% CO_2_ and 95% air at 37 °C for 24 h (1.15 × 107 particles/mL). Then, the cells were washed with sterile PBS twice, fixed in 1% paraformaldehyde solution, incubated with DAPI for 4 min, mounted with Fluoromount-G™ (36313ES60, YEASEN, Shanghai, China), and viewed under a fluorescence microscope. The number of GFP and RFP dots was determined by manual counting of the fluorescent puncta. At least 40 cells were scored in each experiment.

### 2.10. Enzyme-Linked Immunosorbent Assay (ELISA)

YCC cell supernatant samples were collected from different groups, including exosome-, 3-MA-, and RAPA-treated cells. The estradiol (E2) in the cell medium supernatant was evaluated using a bovine E_2_ ELISA kit according to the manufacturer’s instructions (Shanghai Jichun Industrial, Shanghai, China). The minimum detectable concentration was less than 0.1 pg/mL. Briefly, blank (without samples and enzyme standard reagent), control, and experimental groups were set up. After, the samples and HRP-conjugate reagent were added to wells, incubated for 1 h at 37 °C, washed 5 times, the chromogenic reagent was added, and the reaction stopped. Finally, the optical density of each well was measured at 450 nm using an iMark enzyme-labeled instrument (Shanghai Bio-Chain Biological Technology, Shanghai, China).

### 2.11. Statistical Analysis

GraphPad Prism 6.0 software (GraphPad Software, San Diego, CA, USA) was used to analyze the data. All data are presented as means ± standard error of the mean (SEM). A two-tailed Student’s t-test or Bonferroni post-test was applied for comparison of the differences among all data in SPSS 22.0 (IBM, Armonk, NY, USA). Statistical significance was considered at a probability of *p* < 0.05; meanwhile, extremely significant was defined as *p* < 0.01.

## 3. Results

### 3.1. YCC Cultures and Characterization of Yak Follicular Fluid Exosomes

After 48 h of culturing in vitro, the primary YCCs had stable shapes, polygons, and occasional long spindles (Figure 1A). The results of the transmission electron microscopy (TEM) showed that the yak follicular fluid exosomes had a cup base shape with a diameter of approximately 100 nm (Figure 1B) In addition, nanoparticle tracking analysis (NTA) demonstrated that the size of the yak follicular fluid exosomes mainly ranged from 30 to 150 nm and the average was 141.5 nm (Figure 1C). The Western blot assay showed that they could positively express exosome specific protein markers such as CD63 and TSG101 (Figure 1D). These results indicate that the yak follicular fluid exosomes were successfully isolated.

### 3.2. Yak Follicular Fluid Exosomes Promote Autophagy Induction by Being Taken Up by YCCs

Yak follicular fluid exosomes labeled with PKH-26 were incubated with YCCs for 6, 12, and 24 h. PKH-26-labeled exosomes showed a significant increase in fluorescence intensity when compared to the other groups. The results indicate that most of the cells took up the exosomes at 24 h. In contrast, the control group showed no intracellular red fluorescence in vitro. These results suggest that the yak follicular fluid exosomes can be internalized by YCCs (Figure 2). To detect whether autophagy activation occurred in YCCs treated with yak follicular fluid exosomes, we assessed the levels of autophagy-related genes at 6, 12, and 24 h using RT-qPCR. Compared to the control group, the LC3 level increased in the exosome group (Figure 3A1). The same trend was also found for ATG5, Beclin-1, and ATG12 (Figure 3A2–A4), but the variation in the expression of p62 was in the opposite direction (Figure 3A5). The above RT-qPCR data suggest that yak follicular fluid exosomes could upregulate autophagy at the mRNA level.

Further WB results showed that in the exosome groups, protein expression was positively correlated with YCC autophagy, as LC3-II, ATG12-ATG5, and Beclin1 were increased (Figure 3C1,C3,C4). In contrast, P62 expression was decreased (Figure 3C2). Furthermore, YCCs were transfected with RFPGFP-LC3 virus. For the control group, no exosomes were added and the experimental group was cultured with yak follicular fluid exosomes for real-time monitoring of the autophagy flow. RFP was used to label and track LC3, and the weaking of GFP can indicate the fusion of lysosomes and autophagosomes to form autophagosomes. Specifically, the GFP fluorescent protein is sensitive to acidity, whereas the RFP fluorescence signal can persist. In the merged images, yellow dots indicate autophagosomes while free red dots indicate autolysosomes. An increase in both yellow and red dots in YCCs means that the autophagic flux has increased. In this study, we observed more GFP and RFP dots in the yak follicular fluid exosome groups than the control (Figure 4A,B). The quantitative results showed that the yak follicular exosomes also had significantly increased numbers of both yellow (autophagosomes) and red dots (autolysosomes) (Figure 4C). These results indicate much higher amounts of autophagosomes in YCCs treated with yak follicular fluid exosomes. Taken together, these data suggest that the presence of yak follicular fluid exosomes promotes autophagy.

### 3.3. Exosomes Derived from Yak Follicular Fluid Could Increase 2-OHE_2_ Secretion in YCCs

CYP17A1 and CYP19A are essential mammal steroidogenic enzymes. CYP17A1 catalyzes a reaction leading to the formation of androstenedione from pregnenolone while CYP19A catalyzes a reaction leading to the formation of estradiol from androstenedione [24]. CYP1A1 and CYP1B1 thus catalyze two reactions that lead to the formation of 2-OHE_2_ (which is approximately 50% of this process) and 4-OHE_2_ from estradiol. To test whether yak follicular fluid exosomes affect 2-OHE_2_ secretion from YCCs, we observed the expression of CYP19A1 and CYP1A1 using immunofluorescence staining (Figure 5A). We then detected the mRNA expression levels for CYP17A1, CYP19A1, CYP1A1, and CYP1B1 in YCCs and found that CYP17A1, CYP19A1, and CYP1A1 were all increased in the exosome groups when compared with the control (Figure 5B1–B3), but the variation in expression of CYP1B1 was decreased and there was no significant difference between the yak follicular fluid exosomes and control groups (Figure 5B4), suggesting that 2-OHE_2_ secretion may be promoted by yak follicular fluid exosomes. Subsequently, to further confirm these results, WB assays were performed, and the protein expression levels showed the same trend as mRNA (Figure 5C,E). The ELISA results also consistently revealed that estradiol secretion was promoted in yak follicular fluid exosome groups, which was indicated by the upregulation of E_2_ in the YCC supernatant (Figure 5D). The data thus demonstrate that the yak follicular fluid exosome treatment promoted estradiol secretion and the estradiol metabolites were predominately 2-OHE_2_ and not 4-OHE_2_ in YCCs.

### 3.4. Autophagy Is Required for Yak Follicular Fluid Exosome-Mediated 2-OHE_2_ Secretion in YCCs

To investigate whether autophagy regulation was required for 2-OHE_2_ secretion from YCCs, the levels of autophagy-related proteins in YCCs were assessed using WB (Figure 6A,B). The results show that compared with the control group (no exosomes and 3-MA added), the expression levels during autophagy for the positively correlated proteins LC3 and Beclin-1 were increased while the negatively correlated protein P62 was decreased. In contrast, the 3-MA treatment could reverse the effects of the exosomes. The results indicate the efficient inhibition of autophagy by 3-MA. To further confirm that autophagy is required for 2-OHE_2_ secretion in YCCs, ELISA was performed to evaluate the estradiol concentrations in the YCC supernatant. After autophagy was activated by yak follicular fluid exosomes, the concentration of estradiol in the cell supernatant was increased, but the improvement was blocked by autophagy inhibition with 3-MA (Figure 6D). The Western blot results are consistent with the ELISA results, as the expression levels for CYP19A1 and CYP1A1 were decreased (Figure 6E2,E3) while there was no significant difference for CYP17A1 and CYP1A1 in YCCs treated with 3-MA (Figure 6E1,E4). The results confirm that the yak follicular fluid exosomes could active autophagy and subsequently increase the secretion of 2-OHE_2_.

The pharmacological agent RAPA activates autophagy by blocking MAPK/JNK-MTORC1 signaling [25]. To determine whether 2-OHE_2_ secretion is related to autophagy activation, we treated YCCs with different concentrations of rapamycin. The WB results consistently reveal that compared with the control group (no RAPA added), autophagy was promoted by rapamycin-treated YCCs, which is indicated by the upregulation of LC3 and Beclin1 and the downregulation of P62 (Figure 7A,B). Furthermore, the WB results show that the expression levels of CYP19A1 and CYP1A1 were increased whereas CYP17A1 and CYP1B1 were decreased when compared with the control (Figure 7C,D). An ELISA assay was performed to assess the estradiol in the cell supernatant, and the results show that the rapamycin treatment effectively increased estradiol secretion (Figure 7E). Thus, RAPA stimulation induced 2-OHE_2_ secretion from YCCs by activating autophagy. Taken together, the data demonstrate that YCCs treated with yak follicular fluid exosomes showed similar increases in 2-OHE_2_ secretion through the activation of autophagy in cumulus cells treated with Rapa.

## 4. Discussion

Exosomes derived from different cell types have different characteristics and participate in a variety of physiological processes, including autophagy, embryo implantation, placental physiology, semen regulatory function, reproductive biology, and pregnancy [26,27]. Exosomes that have a lipid bilayer membrane can transfer their contents to receptor cells, which is important for intercellular communication [11]. In our study, we found that exosomes derived from yak follicular fluid could enhance autophagy levels in YCCs. This finding is consistent with previous studies showing that exosomes can markedly influence autophagy. Follicular fluid EVs, which fill the ovarian follicle and influence oocyte developmental competency, can originate from the oocyte, cumulus, or mural granulosa cells in the ovarian follicle [28]. Initially, it was reported that huc-MSCs-derived exosomes could activate autophagy via miR-146a-5p/TRAF6 [29]. Breast cancer cell-derived exosomes were also reported to induce autophagy of breast cancer cells by transferring miR-1910-3p [30] while gastric cancer cell-derived exosomes were found to induce the autophagy of neutrophils via HMGB1/TLR4/NF-κB signaling [31]. Moreover, follicular fluid EVs are thought to act as information vehicles during oocyte maturation [32]. The exosomes from the follicular fluid can modulate the arrest of oocyte meiosis and improve the in vitro maturation rate of oocytes, suggesting that follicular fluid exosomes are important for this process [32,33]. Mouse and bovine cumulus–oocyte complexes, via the uptake of bovine follicular fluid-derived exosomes, were found to promote cumulus expansion, which is a critical process for ovulation [34]. Specifically, cumulus expansion was found to be required for meiotic resumption in pig oocytes [35]. Therefore, autophagy may be an effective approach by which to improve mammal assisted reproductive technologies (ARTs). These results combined with our findings further support that exosomes derived from yak follicular fluid could promote autophagy, in YCCs, and this knowledge could lead to new developments to improve ART based on the autophagy pathway.

In the current study, together with the upregulation of four autophagy-related genes, including LC3, Beclin1, ATG5, and ATG12, the reduced P62 levels indicate that yak follicular fluid exosomes enhanced autophagy activity in YCCs, particularly after exosomes were used to treat YCCs for 24 h. Autophagy is a dynamic process regulated by multiple proteins [7,36]. It is activated to maintain cellular homeostasis as cells adapt to different stress situations, such as starvation, aging, nutrient deprivation, and hypoxic stress [37,38]. During the autophagy process, light chain 3 (LC3) exists in autophagosomes and is associated with all types of autophagic membrane. Once autophagy is activated, LC3-I is hydrolyzed into LC3-II through the conjugation of phosphatidylethanolamine, which is correlated with autophagosome formation and elongation [36]. Therefore, the ratio of LC3-II/LC3-I may reflect the level of autophagy [11]. P62 serves as a negative marker of autophagy flux and is inversely correlated with autophagic activity for autophagic flux, during which it interacts directly with LC3 to prevent inclusion body formation [39]. Moreover, autophagosome formation requires two ubiquitin-like conjugation systems [40]. One couples ubiquitin-like MAP1LC3/LC3/Atg8 (microtubule-associated protein 1 light chain 3) to phosphatidylethanolamine, and the other couples ubiquitin-like Atg12 to Atg5 [41]. Therefore, increased levels of Beclin-1 and ATG5-ATG12 conjugate indicate autophagy activation. In contrast, increased levels of P62 indicate inhibition of autophagy. Combining these results with the Ad-mRFP-GFP-LC3 transfection data further indicates that yak follicular fluid exosomes can activate autophagy flux by promoting the degradation of P62 and upregulating of LC3II/I, Beclin-1, and ATG12-ATG5.

Interestingly, we also observed that yak follicular fluid exosome stimulation simultaneously promotes autophagy and increases the secretion of 2-OHE_2_ in YCCs, suggesting that autophagy is probably an essential process during 2-OHE_2_ secretion by YCCs. Moreover, it is worth noting that the expression of the two 2-OHE_2_ synthesis-related proteins, CYP19A1 and CYP1A1, was also remarkably upregulated, and the concentration of estradiol was significantly increased in YCCs after yak follicular fluid exosome exposure. The results indicate that the conversion of androstenedione to estradiol is promoted in YCCs, and that the main metabolite of estradiol is 2-OHE_2_ after yak follicular fluid exosome treatment. The involvement of cumulus cells in the regulation of nuclear and cytoplasmic maturation in mammalian oocytes through paracrine signaling has previously been characterized [36,42]. After the molecular mechanism controlling the autophagic pathway was elucidated, autophagy was observed to be active in cumulus cells. However, the physiological role of autophagy in yak cumulus cells was not previously well understood. Many studies have shown a link between autophagy and steroid hormone production, such as a decrease in autophagy activity in aged rat Leydig cells, and a reduction in sex hormone levels in autophagy-deficient mice with Atg5 expression in the brain [43]. Lysosomal degradation is involved in the regulation of hormone secretion by means of typical autophagy [44]. Steroid hormones such as estradiol and aldosterone are important signal molecules that help to regulate both physiology and growth [45,46]. Estrogen is a hormone that plays an important role not only in the development of female secondary sexual characteristics but also in numerous diseases such as infertility, endometriosis, and other reproductive system-related issues [47,48]. The estrogen estradiol-17β has been shown to mediate the regulation of normal maternal cardiovascular adaptations during pregnancy [49]. A major endogenous estradiol metabolite is 2-OHE_2_, which is a strong antioxidant that accounts for 50% of the estradiol metabolites [50,51]. In the estradiol formation and metabolism processes, cholesterol is first catalyzed to progesterone by cytochrome P450 and 3β-HSD, and then CYP17A1 catalyzes progesterone to androstenedione and testosterone. These are then transformed to estradiol under the actions of CYP19A1 [52]. Ultimately, estradiol is metabolized to 2-OHE_2_ and 4-OHE2 under the action of the rate limiting enzymes CY1A1 and CYP1B1. In this study, we observed a tight association between 2-OHE_2_ secretion and the upregulation of autophagy in YCCs. Importantly, the treatment of YCCs with yak follicular fluid exosomes significantly increased the secretion of 2-OHE_2_. Conversely, the 3-MA treatment reversed this effect, suggesting that the effects of the yak follicular fluid exosomes are associated with autophagy. We also found that RAPA dramatically mimicked the effects of the exosomes. It is thus concluded that autophagy can be used to increase 2-OHE_2_ secretion when cells are treated with yak follicular fluid exosomes.

In this study, yak follicular fluid exosomes were found to activate autophagy and subsequently participate in 2-OHE_2_ production, but the role of 2-OHE_2_ was not entirely elucidated. Previous studies have shown that the cholesterol content in COCs varies during in vitro maturation of porcine oocytes, which affects their ability to be fertilized [53]. However, oocytes were deficient in synthesized cholesterol and required cumulus cells to provide the required products from the cholesterol biosynthetic pathway [54]. Furthermore, cholesterol is a precursor substance for the synthesis of estradiol [55]. There are reports that during in vitro maturation of mouse COCs, 2-OHE_2_ had a beneficial effect on the meiotic and developmental competence of the enclosed oocytes [15]. 2-OHE_2_ was also more effective than other catechol estrogens at inducing oocyte maturation [56]. The data thus indicates that 2-OHE_2_ may be an important link in the hormonal cascade leading to LH stimulation during oocyte maturation [57]. Therefore, we can reasonably suppose that the yak follicular fluid exosomes increase 2-OHE_2_ secretion through the activation of autophagy and that this may be an effective control method by which to improve the embryo quality and pregnancy rate and have a specific effect on the biological functions of COCs. Previous studies have shown that yak follicles were classified as healthy (6–8 mm) or atretic follicles (3–6 mm) upon examination with a stereoscopic microscope [58]. Thus, in our study, we selected widely vascularized follicles larger than 8 mm in diameter with a uniform, bright, and translucent appearance. However, in our study, a limitation is that a stereomicroscope examination of the anatomical follicles was not performed to determine the specific developmental stage of the follicles and to elucidate the effect of 2-OHE_2_ in oocytes; so, we should further explore this in the future.

## 5. Conclusions

In conclusion, this study demonstrates that yak follicular fluid exosomes can activate autophagy in yak cumulus cells, and that the optimal time for yak follicular fluid exosome induction of autophagy in yak cumulus cells is 24 h. In further investigations, we revealed that the yak follicular fluid-derived exosomes can promote the activity of limiting enzymes in estradiol formation and metabolism processes for CYP17A1 and CYP1B1 to increase 2-OHE_2_ secretion via autophagy.

## Figures and Tables

**Figure 1 animals-12-03174-f001:**
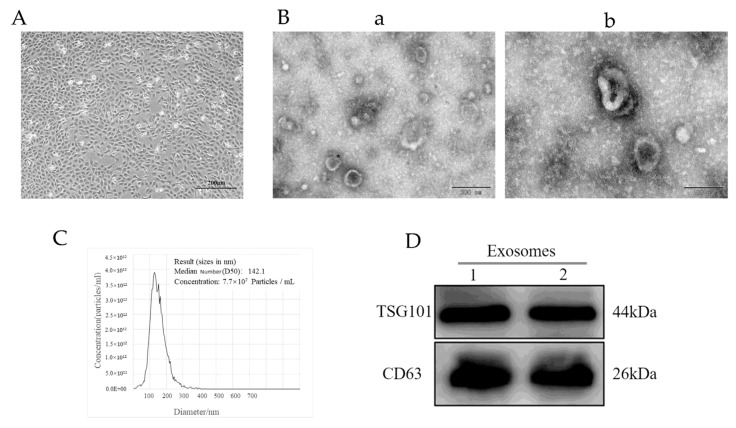
Identification of yak follicular fluid exosomes. (**A**) Phase morphology of isolated YCCs. Scale bar represents 200 μm. (**B**) Size and shape of yak follicular fluid exosomes (white arrows) determined using transmission electron microscopy (TEM). Scale bar represents (**a**) 200 and (**b**) 100 nm. (**C**) Size distribution and concentration of yak follicular fluid exosomes using nanoparticle tracking analysis (NTA). (**D**) Western blot analysis (See Appendix A).

**Figure 2 animals-12-03174-f002:**
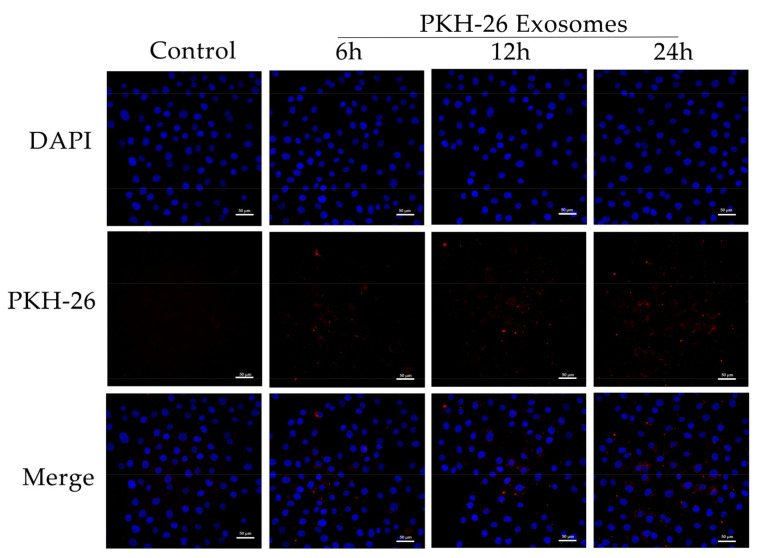
Uptake of yak follicular fluid exosomes by YCCs. Fluorescence microscopy confirmed the location of the yak follicular fluid exosomes that were tagged with PKH-26 (red fluorescence) in YCCs. Scale bar represents 50 μm.

**Figure 3 animals-12-03174-f003:**
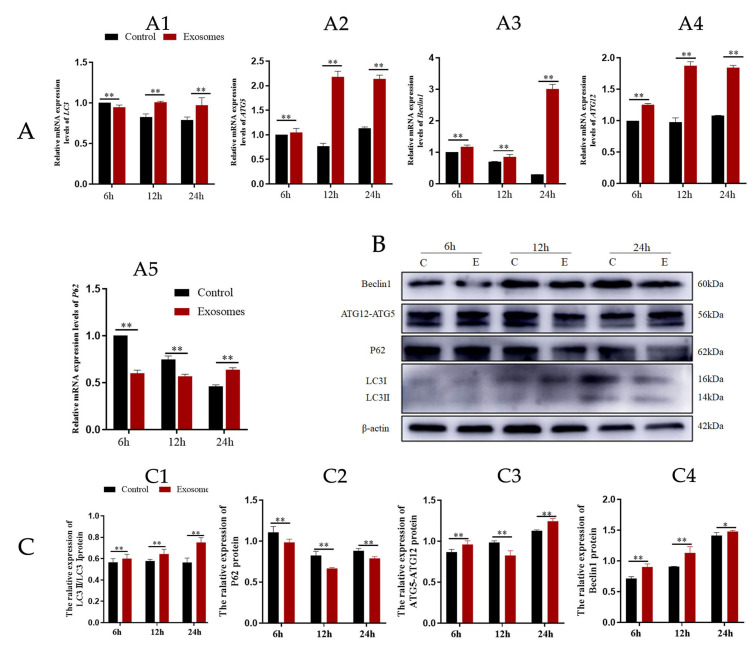
Yak follicular fluid can activate autophagy in YCCs. (**A**) Expression of autophagy-related genes LC3 (**A1**), ATG5 (**A2**), Beclin1 (**A3**), ATG12 (**A4**) and P62 (**A5**) in YCCs using a qPCR assay. (**B**) Western blot analysis of protein expression in YCCs (C: control, E: Yak follicular fluid exosomes) (See Appendix A). (**C**) Quantitative results of Western blot analysis for LC3II/LC3I (**C1**), P62 (**C2**), ATG5-ATG12(**C3**) and Beclin1 (**C4**). Data are expressed as the mean ± SD. *n* = 6. NS means no significant difference. * *p* < 0.05 and ** *p* < 0.01 indicate significant differences between the yak follicular fluid exosome treatments and the control.

**Figure 4 animals-12-03174-f004:**
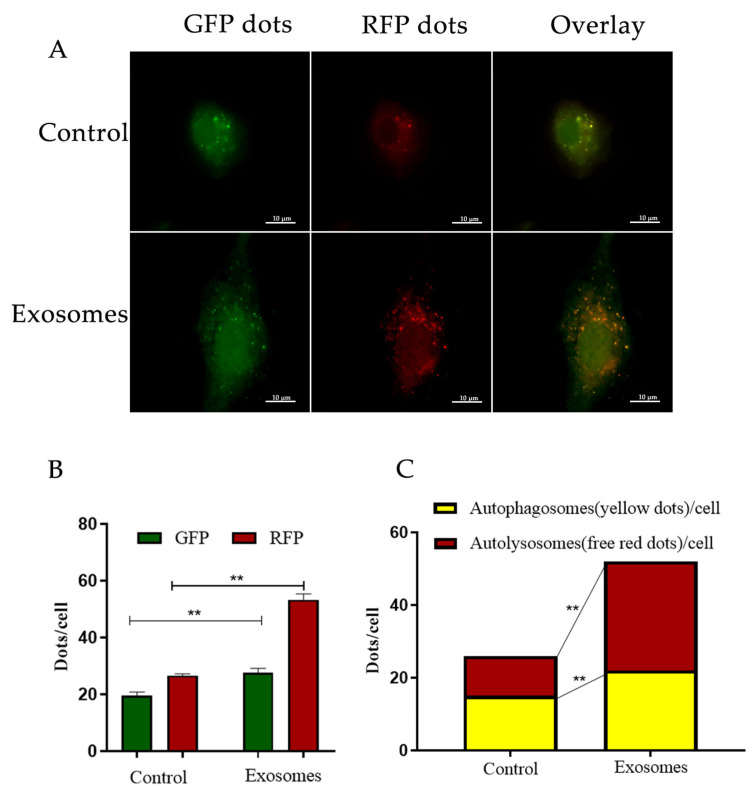
Enhanced autophagy flux of YFFEVs in YCCs. (**A**) Confocal imaging of YCCs with YFFEVs treatments after RFP-GFP-LC3 transfection. Scale bar represents 10 μm. (**B**) Quantitative analysis of GFP and RFP dots. Green dots represent autophagosomes, and red dots represent autophagosomes and autolysosomes. (**C**) The quantitative analysis of merged images with both yellow dots (autophagosomes) and red dots (autolysosomes). Data are expressed as the mean ± SD. *n* = 6. NS means no significant difference. ** *p* < 0.01 indicates a significant difference between the yak follicular fluid exosome treatments and the control.

**Figure 5 animals-12-03174-f005:**
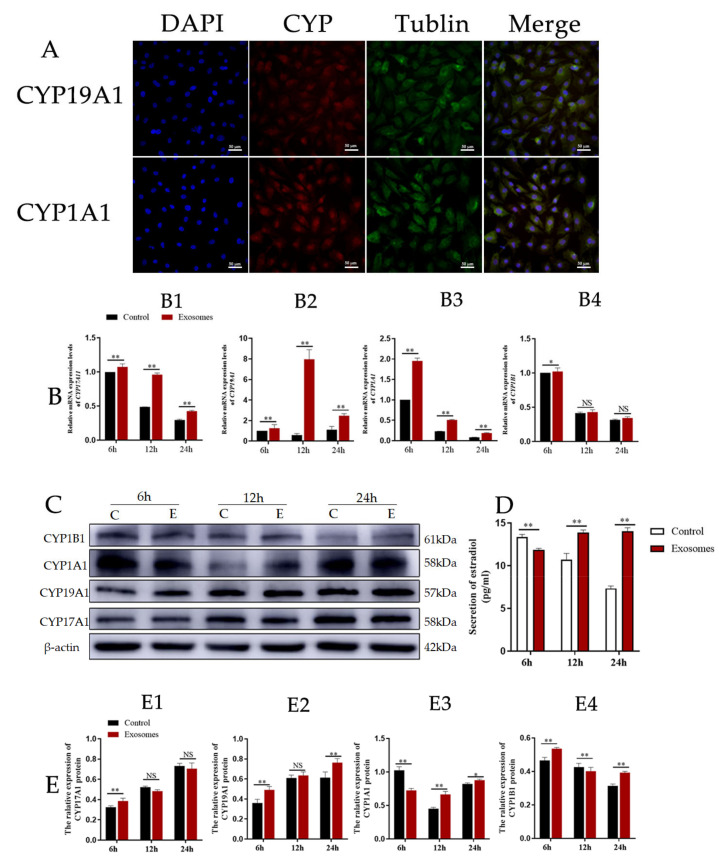
Yak follicular fluid exosomes could increase 2-OHE_2_ secretion in YCCs. (**A**) Immunofluorescence staining of CYP19A1 and CYP1A1 proteins in YCCs. Scale bar represents 50 μm. (**B**) Expression of 2-OHE_2_ secretion-related genes CYP17A1 (**B1**), CYP19A1 (**B2**), CYP1A1 (**B3**) and CYP1B1 (**B4**) in YCCs using the qPCR assay. (**C**) Western blot analysis of the protein expression in YCCs (C: Control, E: Yak follicular fluid exosomes) (See Appendix A). (**D**) ELISA kit analyses of the concentrations of estradiol (pg/mL) in the cell supernatant. (**E**) Quantitative Western blot results for CYP17A1 (**E1**), CYP19A1 (**E2**), CYP1A1 (**E3**) and CYP1B1 (**E4**). Data are expressed as the mean ± SD. *n* = 6. NS means no significant difference. * *p* < 0.05 and ** *p* < 0.01 indicate significant differences between the yak follicular fluid exosome treatments and the control.

**Figure 6 animals-12-03174-f006:**
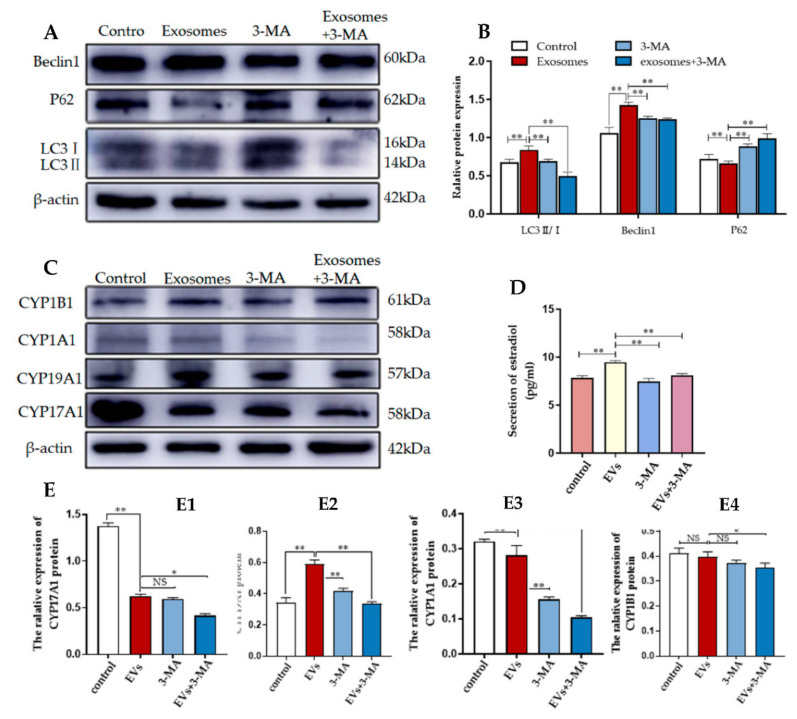
3-MA inhibits autophagy and reduces 2-OHE_2_ secretion in YCCs. (**A**,**B**) Western blot analysis of protein expression in YCCs. (**C**) Western blot analysis of the protein expression in YCCs (See Appendix A). (**D**) ELISA kit analysis of the concentrations of estradiol (pg/mL) in the cell supernatant. (**E**) Quantitative Western blot results for CYP17A1 (**E1**), CYP19A1 (**E2**), CYP1A1 (**E3**) and CYP1B1 (**E4**). Data are expressed as the mean ± SD. *n* = 4. NS means no significant difference. * *p* < 0.05 and ** *p* < 0.01 indicate significant differences between the yak follicular fluid exosome treatments and the control.

**Figure 7 animals-12-03174-f007:**
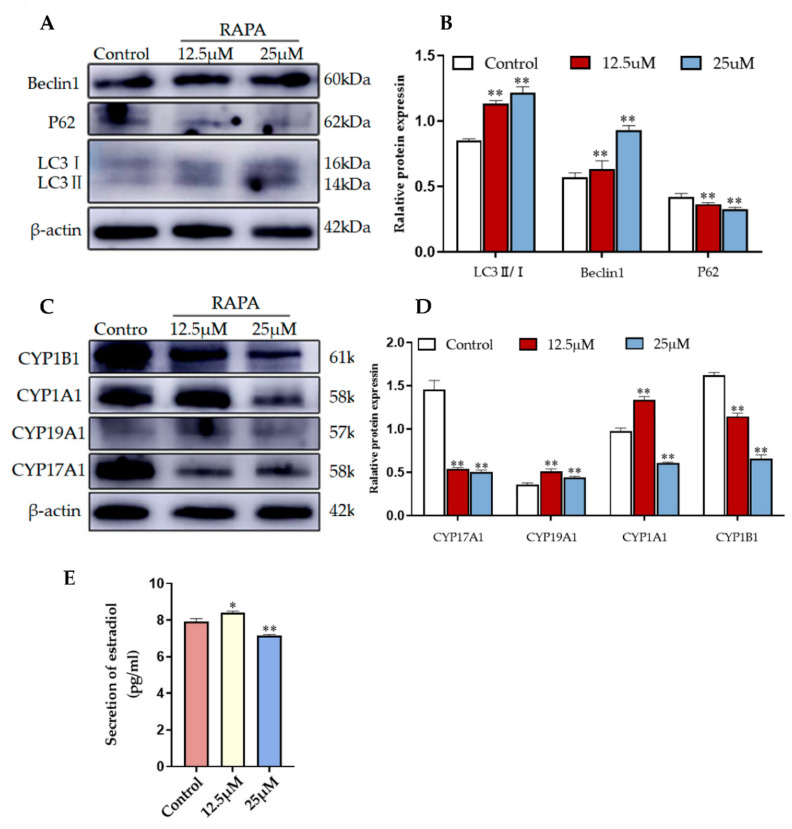
RAPA increases 2-OHE_2_ secretion by upregulating autophagy in YCCs. (**A**,**B**) Western blot analysis of the protein expression in YCCs. (**C**,**D**) Quantitative results of the Western blot (See Appendix A). (**E**) ELISA kit analysis of the concentrations of estradiol (pg/mL) in the cell supernatant. Data are expressed as the mean ± SD. *n* = 3. * *p* < 0.05 and ** *p* < 0.01 indicate significant differences between the yak follicular fluid exosome treatments and the control.

**Table 1 animals-12-03174-t001:** RT-qPCR primers used in this study.

Primer	Forward or Reverse	Sequence (5’–3’)	GenBank No.
LC3B	Forward	AACCAAGCCTTCTTCCTCCT	NM_001001169
	Reverse	ATTGCTGTCCCGAATGTCTC	
BECLIN1	Forward	AGTACCAGCGGGAGTATAGTGA	NM_001033627
	Reverse	CAAGCGACCCAGTCTGAAAT	
ATG5	Forward	ATCAATCGGAAACTCATG	NM_001034579
	Reverse	AGATGTTCACTCAGCCAC	
ATG12	Forward	CCCCTTCTTCTGCTGC	NM_001076982
	Reverse	GGGTCCCAACTTCCTG	
P62	Forward	AGGAGCTTTGGTTCGTGGAA	NM_001205519
	Revers	CCCCTTGACTCTGGCTGTAATA	
CYP17A1	Forward	GCCCAAGACCAAGCACTC	NM_174304
	Reverse	CCCAAACGAAAGGAATAGATG	
CYP19A1	Forward	TGCTGGACACCTCTAACATGC	NM_174305
	Reverse	AAAATCAACTCAGTGGCGAAAT	
CYP1A1	Forward	GTCCCCTTCACCATCCCA	AB060696
	Reverse	CCAAGCCGAAAATAATCACC	
CYP1B1	Forward	GCTTCCGTCTTGGGCTAC	NM_001192294
	Reverse	GGTCAAAGTCCTCTGGGTTC	
β-actin	Forward	ATCGTGCGTGACATCAAAGA	AY141970
	Reverse	CAAGAAGGAAGGCTGGAAAA	

## Data Availability

All data presented in this study are available on request from the corresponding authors.

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
