# Peer review of "Exosomes Derived from Yak Follicular Fluid Increase 2-Hydroxyestradiol Secretion by Activating Autophagy in Cumulus Cells"

_animals, 2022, doi:10.3390/ani12223174_

Round 1

Reviewer 1 Report

In this manuscript, the authors report that autophagy is activated by yak follicular fluid exosomes and increases 2-hydroxyestrodiol secretion in yak cumulus cells. The influence of the unknown factors in the follicular fluid has been estimated to be important, and although the exact mechanism has not been elucidated, its role has been recognized. The elucidation of the role of follicular exosomes in this study will help the understanding of unknown factors in follicular fluid. However, there are some questions about this study.

1. Are follicular exosomes a major regulator of autophagy activation in COC?

2. What happens to the change in COC according to the growth of in vivo follicles?

3. Does it have a positive effect on actual embryo development?

Minor comments

Fig. 1. The figure legends of (B) and (C) have been changed.

Fig. 3. The letters A and C are covering the figure.

Fig. 5. The letter B is covering the figure.

Fig. 6. The letter A, B, C, and D are covering the figure.

Fig. 7 The letter A, B, C, and D are covering the figure.

P. 3 L144 3x106

P. 4 L162 3x105

P. 5 L219 p or P ??

Reviewer 2 Report

The research is meaningful. 

1, 2.1. Preparation of yak ovaries and follicular fluid

The follicular fluid from mature follicles (> 8 mm in diameter) was withdrawn with a syringe and then stored at 80° in a 50 mL centrifuge tube.

Why did you use follicles (> 8 mm in diameter) to get follicular fluid? how do you know the stages of follicles collected from slaughterhouse, Did you really think they are mature follicles? May be they were in growing, in maturating and maturated or in degenerating state. 

2, 2.3. Isolation and culture of yak cumulus cells

After the ovaries were washed three times with sterile saline containing 100 IU/mLpenicillin and 100 mg/mL streptomycin sulfate, cumulus-oocyte complexes (COCs) were aspirated from the antral follicles (8 mm in diameter) [22].

are you sure that the antral follicles used for COCs in your experiment were 8 mm in diameter?

3,3.1. YCC cultures and characterization of yak follicular fluid exosomes

The primary YCCs after 48 h of culturing in vitro had stable shapes, long spindles, and occasional regular polygons (Figure 1A).

It should be changed as : The primary YCCs after 48 h of culturing in vitro had stable shapes, polygons, and occasional long spindles (Figure 1A).

4, Figure 1. Identification of yak follicular fluid exosomes.

 (A) Phase morphology of isolated YCCs. Scale bar represents 200 μm. (B) Size distribution and concentration of yak follicular fluid exosomes using nanoparticle tracking analysis (NTA). (C) Size and shape of yak follicular fluid exosomes(white arrows) determined using transmission electron microscopy (TEM)。

Where are C and white arrows on the pictures?

5, In line 146,For the experimental group, the cells were treated with yak follicular fluid exosomes (120μg/mL).

Why did you use 120μg/mL of exosomes to treat cells, did you check the the size distribution and the exosomes concentration of follicular fluid of follicles in more than 8mm in diameter?

Reviewer 3 Report

Dear authors

The work submitted by Xu et al., I find of very high quality in the development of the ideas, and the development of the experimental part. My suggestions will only go in the way of improving the manuscript and making it easier to read for the community of the area and related sciences.  I consider this manuscript as a real contribution to the specific area in which this group of researchers develops.

The introduction is carried out in an excellent way, exposing the ideas that lead us to formulate the hypothesis or the objective of the work.

The methodology used to demonstrate the hypothesis is accurate and appropriate  to the objectives set. When reading the results section, there is a feeling that a better description of the controls used in each group of experiments is missing.
Statistical analysis section, states the use of a t-test, but this analysis is for the comparison of two experimental groups. For example, Figure 6  when comparing in the figure control, exosomes, 3-MA, and exosomes + 3_MA (Figure 6B, D, E) it is better to use a one-way ANOVA, with a Bonferroni post-test, or Newman-kKeuls, for example. It is not correct to use multiple t-tests in one figure or analysis as you have presented in your manuscript, these are as stated by the authors in the statistical analysis section. You should correct this and see if the statistical significances that you state in the figures and the text are maintained.

The results section is excellent writing, and it helps a lot to follow the authors' idea that they close or conclude at the end of each section the main finding made. But according to what I detailed in the previous paragraph, each figure shows an experiment with different figures or analyses, so the controls or control groups are different. This may lead the reader to confuse the controls used. The recommendation is that the controls or what makes up the so-called control group be made explicit. This would help the readers of the manuscript to concentrate on following the flow of the results and not to be distracted by looking in the manuscript to see which controls were used. Remember that the controls are as important as the experimental groups.

The discussion is well conducted, confronting your results with the existing literature. The writing style of the paragraphs (first exposition of the literature, own results, and finally a conclusion of the topic). In my opinion, in each paragraph I would order it first its results, followed by a comparison with the existing literature, comparing them and making the counterpoints or evidencing the similarities and explanations, to finally give the conclusions of the main idea of the paragraph.

I suggest the authors enter my comments to improve the reading of their work. They should see the statistical analysis, to give more coherence to the manuscript and its results. And from the point of view of style, they take for granted that the reader understands the subject, they should explain to the reader the controls used, and why they investigate the proteins stated in the manuscript so that the reader can follow their main findings.

Reviewer 4 Report

This is a very interesting manuscript worthy of publication because the process of autophagy involving exosomes in follicles may be important for fertilization efficiency.
Please make corrections before accepting the manuscript for publication:
1. conditions for culturing exosomes - atmosphere is not provided, please fulfill description
2. please explain the purpose of passage of exosomes
3. in the test of the manuscript there is no explanation of the abbreviations: 3-MA and RAPA
4.In ELISA description trere is not provided sensitivity, inter- and intraassay.
5. In the statistics description, P description should be corrected.

Round 2

Reviewer 1 Report

Well revised.

Author Response

Dear Reviewer:

Thank you for your comments concerning our manuscript entitled “Exosomes derived from yak follicular fluid increase 2-hydroxyestradiol secretion by activating autophagy in cumulus cells” (animals-1968785). I have uploaded the manuscript after revision, please view it.  If there is still something unreasonable, please let me know in time. I will look forward to it very much.   

Thank you!

Kind regards!